# Realization of Artificial Neurons and Synapses Based on STDP Designed by an MTJ Device

**DOI:** 10.3390/mi14101820

**Published:** 2023-09-23

**Authors:** Manman Wang, Yuhai Yuan, Yanfeng Jiang

**Affiliations:** Department of Electrical Engineering, School of Internet of Things (IoTs), Jiangnan University, Wuxi 214122, China; wmm122122@126.com (M.W.); 6221922015@stu.jiangnan.edu.cn (Y.Y.)

**Keywords:** STT-MTJ, neuron, synapse, image recognition

## Abstract

As the third-generation neural network, the spiking neural network (SNN) has become one of the most promising neuromorphic computing paradigms to mimic brain neural networks over the past decade. The SNN shows many advantages in performing classification and recognition tasks in the artificial intelligence field. In the SNN, the communication between the pre-synapse neuron (PRE) and the post-synapse neuron (POST) is conducted by the synapse. The corresponding synaptic weights are dependent on both the spiking patterns of the PRE and the POST, which are updated by spike-timing-dependent plasticity (STDP) rules. The emergence and growing maturity of spintronic devices present a new approach for constructing the SNN. In the paper, a novel SNN is proposed, in which both the synapse and the neuron are mimicked with the spin transfer torque magnetic tunnel junction (STT-MTJ) device. The synaptic weight is presented by the conductance of the MTJ device. The mapping of the probabilistic spiking nature of the neuron to the stochastic switching behavior of the MTJ with thermal noise is presented based on the stochastic Landau–Lifshitz–Gilbert (LLG) equation. In this way, a simplified SNN is mimicked with the MTJ device. The function of the mimicked SNN is verified by a handwritten digit recognition task based on the MINIST database.

## 1. Introduction

Over the past decade, the spiking neural network (SNN) has become one of the most popular architectures to simulate the brain neural network. Considered as the third-generation neural network, the SNN shows many advantages. The artificial neural network (ANN) is considered as the second-generation neural network. Compared with the ANN, the SNN is more plausible biologically and achieves better performance in pattern recognition tasks [1]. The ANN often uses fairly perfect integrators and a non-linear activation function. The cortical neurons behave as leaky integrations that use conductance-based synapses. Furthermore, the standard training method in the ANN is back propagation, in which each neuron is fed its specific error signal for updating the weight matrix during training. But this kind of learning based on neuron-specific error signals is unlikely to be achieved in the cerebral cortex, where the learning methods are closer to unsupervised learning methods, such as the spike-timing-dependent plasticity (STDP) mechanism [2]. In the SNN, the neural information is stored in the neuron in the form of spike training. When there is an external signal, the neuron is used to integrate the input and the leakage, while the weight of the synapse connecting each neuron is updated based on the STDP mechanism. 

The STDP learning process includes the following stages. The first is the adjustment of the connection strengths (i.e., synaptic weights) based on the relative timing of a particular neuron’s output and input states. The second stage is the hardware implementation of the SNN trained by STDP. In the implementation, the neuron is needed to generate the spiking signal and the synapse for the adjustment of the weight in real time. In the SNN, it shows huge benefits related to its asynchronous processing and massively parallel architecture [1,2]. Recent developments in neuromorphics aim to implement the SNN in hardware to fully exploit its potential in terms of low energy consumption. Nevertheless, the general-purpose computing platforms and the custom hardware architectures implemented using standard CMOS technology cannot rival the power efficiency of the human brain. Hence, there is a need for novel nanoelectronic devices that can efficiently model the neurons and synapses constituting the SNN.

As an emerging non-volatile memory, magnetic random-access memory (MRAM) has many advantages, such as non-volatile data, low power consumption, high integration, strong durability, compatibility with the CMOS process, radiation hardness, etc., and is considered as one of the most promising next-generation memories [3,4,5,6,7,8]. MRAM opens the door to the new computing paradigm, which is different from the traditional Von Neumann architecture. As the core device of MRAM, the magnetic tunnel junction (MTJ) device shows promising properties [9]. At present, it is applied in many fields, including memory, sensors, and neural networks [8,9,10,11,12,13,14].

Current digital implementations of neuromorphic computing rely on large numbers of CMOS transistors, which commonly need a large area and consume a lot of energy. A cutting-edge neuromorphic circuit with a superior architecture is highly needed. For example, the sheer number of synapses for a few-node neural network requires intricate connections and routings, which would be relatively expensive with a CMOS-only solution. The MTJ device can be used to represent the biological neuron and the synapse on a one-to-one basis to mimic the computational dynamics in the human brain [15]. In this sense, the MTJ device offers a compact and energy-efficient solution to take the place of the traditional CMOS-based neural network [16,17].

In the paper, a dynamic model of the MTJ device is established first. Based on the operation mechanism of the MTJ device, a high resemblance is shown between the magnetization dynamics of the MTJ device and the STDP mechanism observed in biological synapses. Also, there exists a high resemblance between the magnetization dynamics of the MTJ device and the characteristics observed in biological neurons. Finally, a demo SNN based on the MTJ synapse and the MTJ neuron is constructed, which is used to solve the image recognition problem in the paper. Compared with other works, a different neuron structure is adopted in the paper. The detailed explanation of how to convert pixel information into presynaptic spikes is shown. The corresponding neuron reset circuit is designed to implement STDP in hardware.

The rest of the paper is organized as follows. Section 2 introduces the MTJ model strategies. Section 3 presents the similar characteristics between the MTJ device and the biological synapse. Section 4 presents the similar characteristics between the MTJ device and the biological neuron. In Section 5, the SNN is mimicked by the MTJ device and is applied to distinguish the two types of handwritten digit images. Finally, conclusions are made in Section 6.

## 2. Dynamic Model of the STT-MTJ Device for Simulation

The model of the STT-MTJ device was constructed based on the Landau–Lifshitz–Gilbert (LLG) equation [15,16,17]:(1)$$\frac{d\vec{m}}{dt}=-\gamma \vec{m}\times {\vec{H}}_{eff}+\alpha \vec{m}\times \frac{d\vec{m}}{dt}-K_{STT}J_{STT}\vec{m}\times \left(\vec{m}\times {\vec{m}}_{p}\right)$$
where $\vec{m}$ is the unit magnetic vector of the free layer; ${\vec{m}}_{p}$ is the unit magnetic vector of the pinned layer; *J_STT_* is the current density of the MTJ device and *J_STT_* = *I_STT_*/Area; ${\vec{H}}_{eff}$ is the effective magnetic field, including the demagnetization field; and $K_{STT}=\mu_{0}\gamma P\hbar /2et_{FL}$ is the STT term. Other parameters are listed in Table 1 [18].

Considering the probabilistic switching behavior of the MTJ device, two stochastic aspects are included in the MTJ model. The first one is the angle between the stochastic initial magnetization vector and the easy axis. The second aspect is the stochastic thermal fluctuation field caused by thermal noise. For these two additional stochastic effects of the MTJ device, two corresponding stochastic terms are included in the LLGS equation to simulate the influence on the probabilistic characteristics of the MTJ device.

To simulate the first stochastic term, the initial value of $\vec{m}$ is set with polar coordinates as follows:(2)$${\vec{m}}_{0}=\left(\sin\theta_{0}\cos\varphi_{0},\sin\theta_{0}\sin\varphi_{0},\cos\theta_{0}\right)$$

In most cases, $\varphi_{0}=0$. $\theta_{0}$ is the initial angle, which follows a Gaussian distribution as follows [19]:(3)$$\theta_{0}∼N\left({\bar{\theta }}_{0},\sqrt{k_{B}T/\mu_{0}M_{S}H_{K}V_{FL}}\right)$$
where $\varphi_{0}=0$ and ${\bar{\theta }}_{0}=\sqrt{1/2\Delta }$ is the average value of $\theta_{0}$ [20,21,22].

The other stochastic term is the random thermal fluctuation field, ${\vec{H}}_{f}=\left[H_{fx,}H_{fy,}H_{fz,}\right]$. The three components of ${\vec{H}}_{f}$ in the *x*, *y*, and *z* directions follow a Gaussian distribution as follows [23]:(4)$$H_{fx}{}_{,fy,fz}∼N\left(0,\frac{1}{\mu_{0}}\sqrt{\frac{2\alpha k_{B}T}{\gamma M_{S}V_{FL}\Delta t}}\right)$$

Δt is the time step of the simulation. The LLGS equation with stochastic terms is generally named the SLLGS equation.

The typical sandwich structure of the STT-MTJ device is shown in Figure 1a, including the free layer, oxide layer, and reference layer, respectively [3]. The magnetization direction of the reference layer is fixed at (0, 0, 1), and the magnetization of the free layer can store information. As shown in Figure 1, with $\theta_{0}$ = 5°, when *I_STT_* = 5 μA, the STT moment is not large enough and $\vec{m}$ remains at (0, 0, 1) due to the damping term. At this time, the MTJ device is in the P state. When *I_STT_* = 200 μA, the MTJ device is switched to the AP state. At this time, $\vec{m}$ is changed to be (0, 0, −1).

## 3. Design of the STT-MTJ-Based Synapse

There exists a high resemblance between the STT-MTJ device and the biological synapse. In biology, the synapse acts as the bridge between the neurons. The neuron emitting a signal is called the presynaptic neuron (PRE), and the neuron receiving the signal is called the post neuron (POST). The synapse is used to connect the PRE with the POST. Based on the STDP mechanism, the update of the synaptic weight depends on the spiking time modes of the PRE and the POST. If the PRE spike is ahead of the POST, the synaptic weight will be increased. On the contrary, if the PRE spike lags after the POST, the synaptic weight will be decreased accordingly. The mathematical expression of the STDP mechanism is as follows [2]:(5)$$\left\{\begin{array}{l} \Delta w=A_{+}\exp(\frac{-\Delta t}{\tau_{+}}),\Delta t>0\\ \Delta w=-A_{-}\exp(\frac{\Delta t}{\tau_{-}}),\Delta t<0 \end{array}\right.$$
where Δw is the relative change of the synaptic weight and *A*_+,_ *A*_−,_ *τ*_+,_ and *τ*_−_ are the constants.

Δt is the time difference between the PRE and the POST spikes. $\Delta t=t_{PRE}-t_{POST}$, and *t_PRE_* is the moment when the PRE is activated, while *t_POST_* is the moment when the POST is activated.

The synaptic weight is increased with the positive time windows (with Δt > 0); this is called long-term potentiation (LTP). For the negative time windows (with Δt < 0), the synaptic weight is decreased, which is called long-term depression (LTD). According to the STDP mechanism, the synaptic weight can be programmed in situ based on the spiking timing information transmitted between the spiking neurons.

A similar adjustment mechanism is also observed in the MTJ device in terms of the device conductance. For the perpendicular MTJ device (P-MTJ), its conductance can be adjusted by controlling the pulse width of the voltage. With the positive write current flowing from the free layer to the pinned layer, the resistance of the MTJ is increased. On the contrary, with the negative write current flowing from the pinned layer to the free layer, the resistance is decreased.

Figure 2 shows the designed synapse based on the STT-MTJ device. The structure is shown in Figure 2a. The 1T-1MTJ cell is used as the synaptic connection between the PRE and the POST. The gate voltage of the NMOS, *V_G_*, is controlled by the membrane potential of the PRE, while the node voltage at the top of the MTJ, *V_T_*, is controlled by the membrane potential of the POST. Figure 2b shows the schematic of the time sequences of *V_G_* and *V_T_* under the 1T-1MTJ structure, which are controlled by the PRE and the POST, separately. When Δ*t* > 0, *V_T_*_+_ is overlapped with *V_G_*. In this condition, the internal current flows from the fixed layer to the free layer in the MTJ device, driving the MTJ to be switched to the P state. So, the conductance of the MTJ device is increased.

On the contrary, when Δ*t* < 0, *V_T_*_−_ is overlapped with *V_G_*. The current in the MTJ device flows from the free layer to the fixed layer, driving the MTJ to the AP state. So, the conductance of the MTJ device is decreased.

Since the MTJ device in the AP state has a low conductance, while the MTJ in the P state has a high conductance, the MTJ device in the AP state is used to mimic the synapse with a weight of ‘zero’, and the P state is used to mimic the synapse with a weight of ‘one’. Based on the designed structure in Figure 2, the MTJ device can be used to simulate the STDP mechanism.

Next, the behavior of the MTJ synapses was investigated with the implementation of the handwritten digit images in the MNIST database [24,25]. Figure 3 shows one of the images used in the paper. As shown in Figure 3a, the image of the handwritten digit “4” is a 28 × 28 matrix, with 784 pixels in total. In the SNN field, the image is transformed into a current pulse sequence, which is named a presynaptic pulse sequence. During the changing process, the basic principle is that the pixel in the pure black area is noted as ‘0′, while the pixel in the pure white area is noted as ‘1′. As show in Figure 3b, these pixels are converted to a series of current pulses, where the pixels close to ‘0′ are converted into a negative current pulse and the pixels close to ‘1′ are converted into a positive current pulse.

For the generated presynaptic pulse sequence in Figure 3b, the reconstructed synapse is shown in Figure 4b. It can be seen that the reconstructed image is hard to read. The 784-pixel information in the image is transformed into the corresponding 784 current pulse sequences (also known as presynaptic pulse sequences). As shown in Figure 4b, the 784 random magnetic vector angles are generated based on the uniform distribution, θ. This indicates that the initial state of the MTJ synapse is random, making it difficult to distinguish the image’s content. To improve its quality, repetitive training is needed. As shown in Figure 4c, 10 training steps were conducted on the 784 synapses, i.e., with repeated writing of the presynaptic sequences 10 times. It can be seen that the results gradually tended to stabilize, with the numbers gradually becoming clear and readable. In the corresponding handwritten digital image, the pixels in the black area are close to zero, with the MTJ device in the AP state and a weight of 1. On the contrary, the pixels in the white area correspond to the MTJ device in the P state and a weight of 0.

## 4. STT-MTJ-Based Neuron

The similar properties of the STT-MTJ device and the biological neuron are studied in this section. Figure 5 shows a schematic diagram of the membrane potential of a biological neuron, in which the spike and the leakage of the input are integrated together. The neuron would be activated when the membrane potential exceeds the threshold voltage [7].

The similar characteristics of the MTJ device were shown by micromagnetic simulation based on the MTJ model in Section 2. The stochastic magnetic simulation was carried out based on the P-MTJ device with the ${\vec{m}}_{p}$ = (0, 0, −1). The states of the MTJ in the magnetic dynamic model can be characterized by the m_z_, which is the z-component of $\vec{m}$, with *m_z_* = cosθ. The term m_z_ can be used to stand for the membrane potential of the biological neuron in the STT-MTJ-based SNN structure.

The first two terms on the right-hand side of the LLGS equation described in Equation (1) are related to the leakage ones of the membrane potential in the magnetization dynamics, while the last term is related to the input pulse applied on the MTJ neuron. Figure 6 shows the integration process and the activation process of the m_z_ in the STT-MTJ device. As shown in Figure 6, the input pulse with a 1 ns period and a 0.55 ns pulse width is adopted as the input pulse signal of the neuron. The precession of *m_z_* is simulated and four periods are shown in Figure 6a. It can be seen that *m_z_* can exhibit the integrated function, showing the accumulation effect of the multiple inputs and the leakages. Based on the integration function, the pulse has an obvious influence on the value of m_z_, with *m_z_* being increased with a pulse and decreased without a pulse.

The activation of the neuron occurs when the membrane potential exceeds the threshold. In the MTJ neuron, the activation corresponds to the switching behavior of the MTJ device and is also presented by *m_z_*. As shown in Figure 6b, *m_z_* is switched from −1 to +1 successfully with six-cycle pulses as the input, which means that the MTJ neuron can be activated successfully. Due to the non-volatile property of the STT-MTJ device, m_z_ can be kept at +1 even without the input pulse. Therefore, a reset circuit must be designed to reset the activated MTJ neuron. The reset period is similar to the refractory period observed in the biological neuron. The reset neuron cannot be activated again for a short time after being activated.

The operation of the MTJ neurons can be divided into three stages, namely, the write stage, the read stage, and the reset stage. As shown in Figure 7, in the writing stage, *V_WRITE_* is high. The input synaptic current, I, is transmitted through the heavy metal layer. The MTJ neuron is driven by the input current. The state of the MTJ device is switched from the P state to the AP state. So, the neuron is activated. In the reading phase, *V_READ_* is high, and the state of the MTJ device is determined by the node voltage, *V_SPIKE_*, between the reference MTJ and the MTJ neuron. The read *V_SPIKE_* corresponds to a low-level MTJ in the P state, while the read *V_SPIKE_* corresponds to a high-level MTJ in the AP state. In the reset stage, if *V_SPIKE_* is high, a reset operation is initiated. Reverse current flows through the heavy metal layer, causing the MTJ neuron to be switched from the AP state to the P state, terminating the activation state.

Besides the integration of the input and the leakage, probabilistic activation is another characteristic of the biological neuron. The probability of neuron dynamics [26] mainly comes from the randomness of ion channel switching and the randomness of neurotransmitter releasing. The switching behavior of the MTJ device is also probabilistic in nature. The switching probability of the MTJ (from AP to P or P to AP) is increased with the magnitude of the input current. Therefore, the switching probability of the MTJ device can be mapped with the activation probability of the biological neuron [27]. The activation probability of the biological neuron typically varies non-linearly with the input synaptic current [2,28], which is similar to the non-linear variation of the switching probability of the MTJ device with the applied current.

The switching probability of the MTJ neuron can be adjusted by many factors. As shown in Figure 8, the switching probability can be changed with the variations of the MTJ free layer thickness, *t_FL_*, and the duration of the input current, *t_pw_*. The applied input current, *I*, is a square wave signal with *t_pw_* width. It can be seen that the switching probability of the MTJ device is decreased with increasing *t_FL_* (as shown in Figure 8a), while it is decreased with decreasing *t_pw_* (as shown in Figure 8b). By controlling *t_FL_*, *t_pw_*, or other factors, the activation function of the MTJ neuron can be adjusted by the changing of the switching probability. So, the MTJ neuron can be designed to be sensitive to specific inputs and to be inactivated with other inputs.

## 5. MTJ Mimics the SNN with Application in Image Recognition

Figure 9 shows the application scenario of the SNN for handwritten digit image recognition. Only the connections for one neuron are shown in the illustration in Figure 9. 

The information of the input handwritten digit images is transferred to neurons through synapses. The neurons receive the postsynaptic current pulses. The synapses and the excitatory neurons are mimicked by the MTJ devices, as introduced in Section 3 and Section 4. The role of the inhibitory neuron is equivalent to the peripheral reset circuit, which prevents the neuron from being activated repetitively within a short limited period of time, simulating the refractory period of the biological neurons.

Figure 10 shows ten images of handwritten digits, including five images of “1” and five images of “0”. The images were used as the input samples for the MTJ-based image recognition.

Figure 11 and Figure 12 show the recognition processes for the handwritten digits “1” and “0” based on the MTJ synapses and the MTJ neuron, respectively. Each set of figures includes the original image, the random initial synapses with the training process for the synapses repeated ten times, the post-synapse current pulse, and the *m_z_* of the MTJ neurons. The same MTJ neurons are used in Figure 11 and Figure 12. The switching probability function of the MTJ neuron is adjusted so that it is sensitive to the input of the handwritten digit “0”, while it is not activated when the images of the handwritten digit “1” are inputted. The recognition function is achieved.

## 6. Conclusions

Neuroscience verifies that the signals in the brain are transmitted in the form of spikes between neurons via synapses. The SNN shows a more biologically realistic perspective and has become the most popular computing model for implementing low-power and high-accuracy recognition. In the paper, a novel SNN was designed based on the STT-MTJ device to differentiate images of handwritten digits in the MINIST database. Based on the micromagnetic simulation of the STT-MTJ device, both the similar characteristics of the MTJ device and biological synapses, as well as the similar characteristics between the MTJ device and biological neurons, were presented. By using the designed MTJ-based synapse and the MTJ-based neuron for image recognition, the average accuracy can reach up to 95%, which is very close to the typical SNN network accuracy. Owing to the characteristics of the MTJ device, the memory power consumption in standby mode is almost zero, which is better than the CMOS device. The proposed MTJ-based neural network shows remarkably promising power and area efficiency. This is a promising candidate for a neural network, with the benefits of easy manipulation, low energy consumption, and high accuracy.

## Figures and Tables

**Figure 1 micromachines-14-01820-f001:**
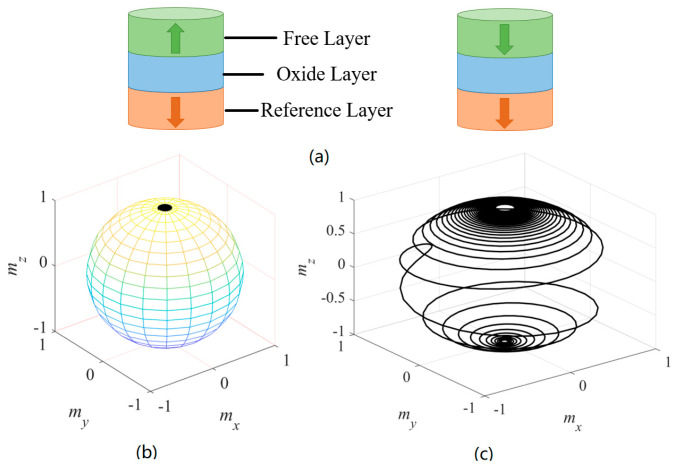
(**a**) The typical sandwich structure of the MTJ device. The left MTJ device is in the anti-parallel (AP) state, while the right MTJ device is in the parallel (P) state. (**b**) With the values of *m* = (sin 5°, 0, cos 5°) and *I_STT_* = 5 μA, the spin transfer torque is not large enough to switch the device because of the damping effect. At this time, *m* = (0, 0, 1), the MTJ device is in the P state. (**c**) With the values of *m* = (sin 5°, 0, cos 5°) and *I_STT_* = 200 μA, the MTJ device is switched to the AP state at this time and *m* = (0, 0, −1).

**Figure 2 micromachines-14-01820-f002:**
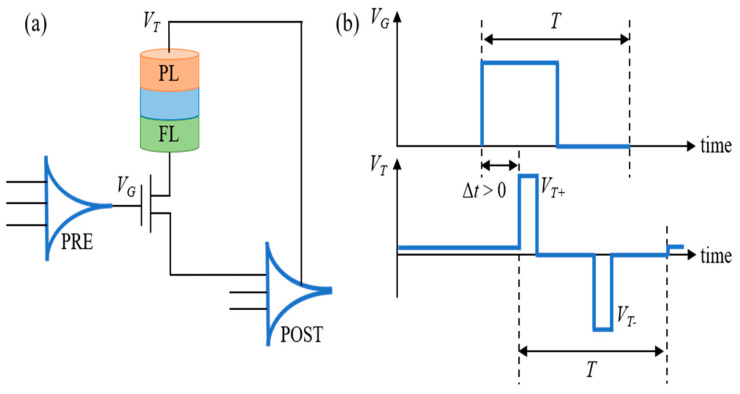
The designed synapse based on the STT-MTJ device. (**a**) The 1T-1MTJ device can be used to mimic the synapse. The NMOS transistor is used as the communication between the PRE and the POST. (**b**) The STDP-based pulsed signal. The time information of the *V_G_* pulse is controlled by the membrane potential of the PRE. The *V_T_* pulse is controlled by the membrane potential of the POST. The pulse timing information determines the positive and negative of the time window, which in turn determines the current of the MTJ device, resulting in a corresponding change in the conductance of the MTJ device.

**Figure 3 micromachines-14-01820-f003:**
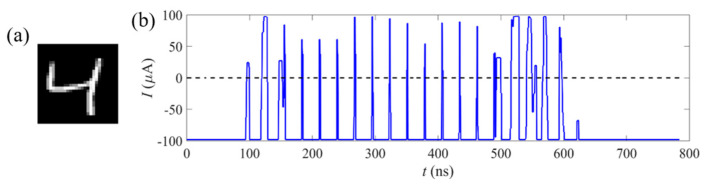
(**a**) The image of the handwritten digit “4”. (**b**) The pixels of the handwritten digit are converted to the current pulse sequence, which is named the presynaptic pulse sequence. The pixels close to ‘0’ are converted into a negative current pulse, and pixels close to ‘1’ are converted into a positive current pulse.

**Figure 4 micromachines-14-01820-f004:**
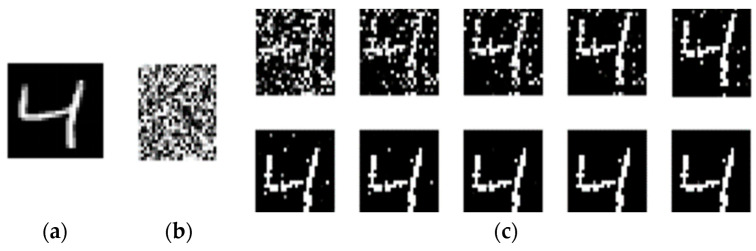
(**a**) The original handwritten digit “4”. (**b**) The reconstructed synapse. (**c**) The input process was repeated 10 times for training synapses.

**Figure 5 micromachines-14-01820-f005:**
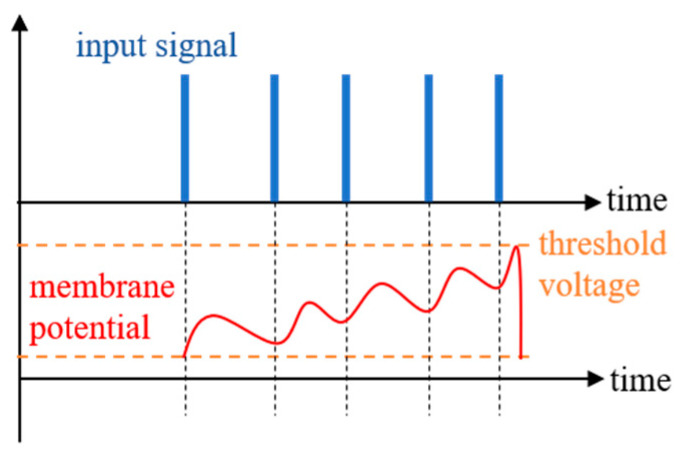
Schematic diagram of the membrane potential of a biological neuron.

**Figure 6 micromachines-14-01820-f006:**
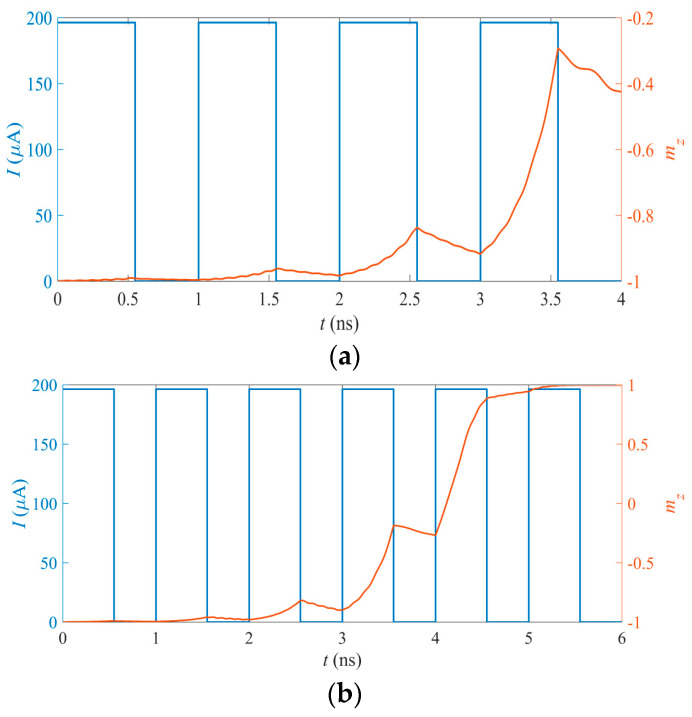
(**a**) The integration function of *m_z_* in the STT-MTJ device. *m_z_* shows the precession tendency due to the application of the input pulses. The precession starts due to the applied pulse and then starts to leak when the pulse is removed. (**b**) The activation function of *m_z_* in the STT-MTJ device. *m_z_* precesses due to the application of the input pulses. The precession starts due to the applied pulse and then starts to leak after the pulse is removed. After *m_z_* reaches the threshold, it remains in the activated state due to the non-volatility of the MTJ device.

**Figure 7 micromachines-14-01820-f007:**
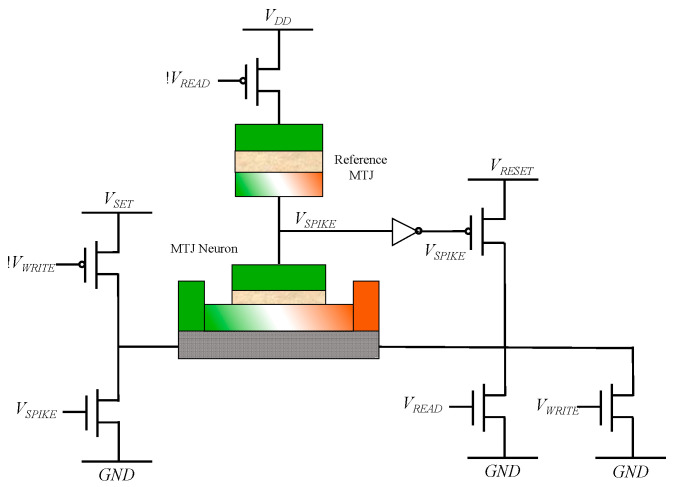
Reset circuit of the MTJ neuron. When the MTJ neuron is activated, the *V_SPIKE_* is high during the reading period, initiating a reset. The reverse current flows through the heavy metal layer, resetting the MTJ neuron back to the P state.

**Figure 8 micromachines-14-01820-f008:**
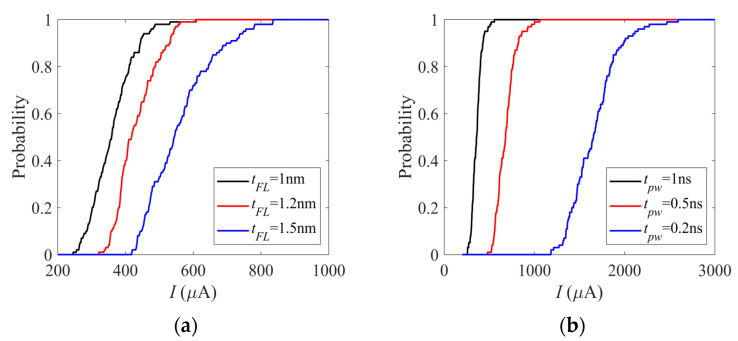
The switching probabilities of the MTJ device are varied non-linearly with the input current, *I*. The probabilistic switching can be directly mapped with the stochastic activation properties of the neuron. (**a**) The switching probability of the MTJ device is decreased with increasing *t_FL_*, while *t_pw_* is kept as 1 ns. (**b**) The switching probability of the MTJ device is decreased when *t_pw_* decreases, where *t_FL_* is kept as 1 nm. It should be noted that the 0.2 ns pulse width is very short, so a large current is needed for certain switching.

**Figure 9 micromachines-14-01820-f009:**
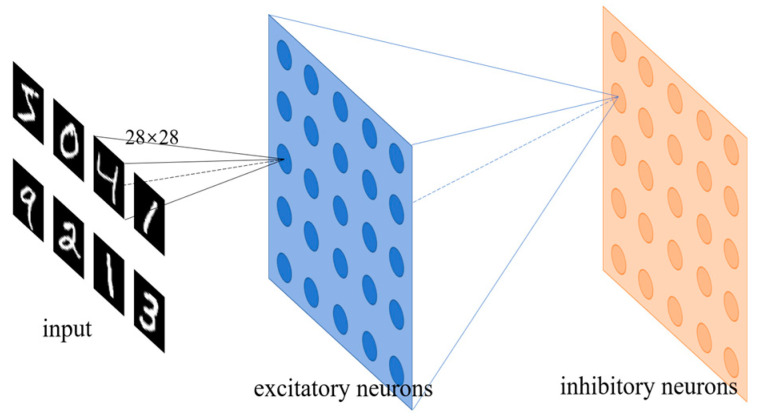
Schematic of the SNN for handwritten digit image recognition. The input includes the pixel information of the handwritten digit image.

**Figure 10 micromachines-14-01820-f010:**

Ten images of the handwritten digits “0” and “1” used for the MTJ-based image recognition.

**Figure 11 micromachines-14-01820-f011:**
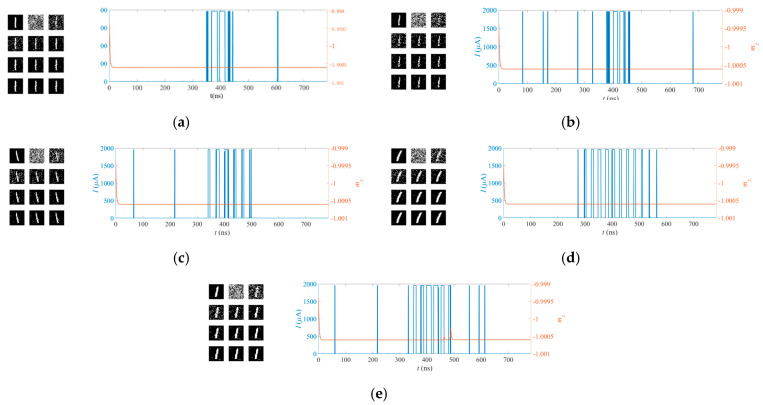
The recognition process for the handwritten digit “1” based on the MTJ synapses and the MTJ neuron. According (**a**–**e**), it can be seen that the input of the handwritten digit “1” does not activate the MTJ neuron. Each set of figures includes the original image, the random initial synapses, the training process for the synapses repeated ten times, the post-synapse current pulse, and the *m_z_* of the MTJ neurons. Different columns correspond to the samples shown in Figure 10.

**Figure 12 micromachines-14-01820-f012:**
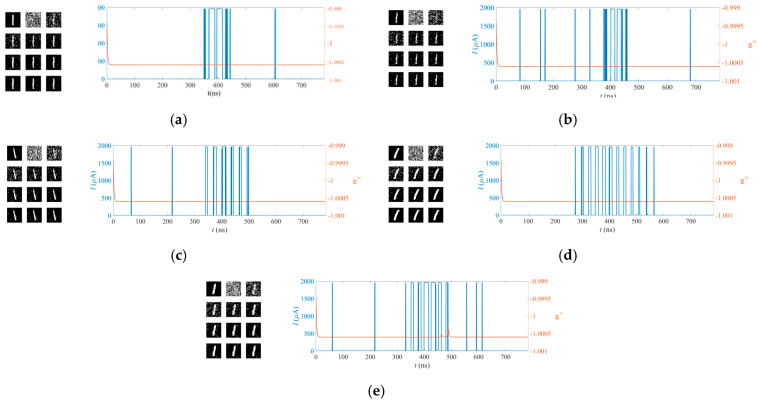
The recognition process for the handwritten digit “0” based on the MTJ synapses and the MTJ neuron. According (**a**–**e**), it can be seen that the input of the handwritten digit “0” activates the MTJ neuron. Each set of figures includes the original image, the random initial synapses, the training process for the synapses repeated ten times, the post-synapse current pulse, and the *m_z_* of the MTJ neurons. Different columns correspond to the samples shown in Figure 10.

**Table 1 micromachines-14-01820-t001:** Parameters.

Parameter	Description	Value
*t_FL_*	Thickness of free layer	1.0 nm
*d*	Diameter of free layer	50 nm
*Area*	Area of free layer	π *d*^2^/4 nm^2^
*P*	Spin polarization of STT	0.4
*μ* _0_	Permeability in free space	$4\pi \times 10^{-7}$ H/m
*H_eff_*	Effective anisotropy field	0.4/*μ*_0_ A/m
*M_S_*	Saturation magnetization	1.0 × 10^6^ A/m
*α*	Magnetic damping constant	0.0127
*γ*	Gyromagnetic constant	1.76 × 10^11^ s^−1^T^−1^
*k_B_*	Boltzmann constant	1.38 × 10^−23^ J/K
*T*	Temperature	300 K
*TMR(0)*	TMR ratio for 0 bias	1.8

## Data Availability

All data generated or analyzed during this study are included in the article.

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
