# Peer review of "Realization of Artificial Neurons and Synapses Based on STDP Designed by an MTJ Device"

_micromachines, 2023, doi:10.3390/mi14101820_

Round 1
Reviewer 1 Report
In this paper, the authors have presented an interesting design of SNN based on STT-MTJ devices. The contents are very interesting and useful results. But, in order to appeal them to the readers, I would like to give the following comments.
(1) Title: The title is too abstract to understand the contents of this paper. I recommend you to revised it so that the contents can be guess. The keyword is: realization of both neurons and synapses. STDP, etc.
(2) Page 3: You wrote "As shown in Fig. 1, assume theta=5 degree, when ISTT = 5 μA, the STT moment is not large enough, m switches to (0, 0, 1) due to damping term, the MTJ is at P state at this time". But, I cannot find that from Fig. 1. So, this is because the explanation of Fig. 1 is not sufficient. Also in the caption of Fig. 1, "(b) m = (sin 5°, 0, cos 5°)" is insufficient. Usually, the caption should be "XXX for m = (sin 5°, 0, cos 5°)". Please consider it. Also for (c).
(3) Fig. 2: During the training and inference, the same configuration is used as that written in Fig. 2 ? If so, write so, and if the different configurations are switched, please show both.
(4) Fig. 3: You wrote "the pixels close to ‘0’ are converted into a negative current pulse, and pixels close to ‘1’ are converted into a positive current pulse", which means that the voltage is binary ? But, in Fig. 3, analog voltage is applied. Please add the explanation to convince the readers.
(5) Fig. 4: I cannot understand the meaning of each figure in Fig. 4. Please explain the details.
(6) Fig. 6: mz is output of the neuron ? So, how can you convert mz to spiking pulses ?
(7) Fig. 6 and Fig. 7 is very similar. So, how about merge them to one figure and summarize the explanation ?
(8) Fig. 8: What is the "switching probability" ? Please define it.
(9) Fig. 9: Could you show the circuit schematics for the entire SNN ? It helps the readers to understand your system. Moreover, I cannot understand "Only the connections for one neuron are shown for illustration in Fig. 9".
(10) Fig. 11 and Fig. 12: Why 3x4=12 image patterns are shown for each figures ? I imagine that one image pattern is inputted and one spike signal series is outputted. Please explain.
(11) I cannot understand how you trained this SNN with STDP. MNIST is a labeled database, so usually supervised learning is done. On the other hand, STDP is a method for unsupervised learning. So, I wonder how you do it ?
Author Response
Thank you for your suggestions, here are answers for your review.Please see the attachment.

Reviewer 2 Report
The author proposes a novel SNN paradigms based STT-MTJ device, whose performances are simulated by LGG equations. The STDP rules are achieved by using 1T1MTJ synapse. This paper is well organized and written, which can be accepted after considering the following points.
1. A comparison with other relative work are required to stress the main contribution and merits of this work.
2. Fig. 4, Fig.10 and Fig.11 are required to re-plot and re-organize, which are hard to follow for readers.
3. By using the common MNIST, the accuracy should be provided. Typical STDP results are also needed.
That is OK.
Author Response

(The authors gave the same response as above.)

Round 2
Reviewer 1 Report
The author have responded most of my last comments properly. I will give the following minor comments.
3) The manuscript is the same ? The red marker part is wrong part ? You wrote "We have made corrections". So, please check again.
10) Please indicate which is the original handwritten digital, initialized, ... ?
11) I cannot understand yet. I can understand that the classification can be done without labels. But, if you don't use label, how the recognition of each number (namely 0, 1, ...) can be done ?
